# Robust Secure Resource Allocation for RIS-Aided SWIPT Communication Systems

**DOI:** 10.3390/s22218274

**Published:** 2022-10-28

**Authors:** Bencheng Yu, Zihui Ren, Shoufeng Tang

**Affiliations:** 1School of Information and Control Engineering, China University of Mining and Technology, Xuzhou 221116, China; 2Xuzhou College of Industrial Technology, Xuzhou 221140, China

**Keywords:** channel uncertainty, information leakage, harvested energy, RIS

## Abstract

Aiming at the influence of channel uncertainty, user information leakage and harvested energy improvement, this paper proposes a robust resource allocation algorithm for reconfigurable intelligent reflector (RIS) multiple-input single-output systems based on imperfect channel state information. First, considering the legal user minimum secret rate constraint, the base station maximum transmit power constraint and the RIS phase shift constraint with the bounded channel uncertainty, a joint optimization of the base station active beam, energy beam and RIS phase shift is established. A multivariate coupled nonlinear resource allocation problem for matrices is addressed. Then, using S-procedure and alternating optimization methods, the original non-convex problem is transformed into a deterministic convex optimization problem and an alternating optimization algorithm based on continuous convex approximation is proposed. The simulation results show that the proposed algorithm has better fairness harvested energy compared with the traditional robust algorithm.

## 1. Introduction

With the development and popularization of Internet of Things technology, the rapid increase in the number of wireless devices and data traffic brings huge energy consumption [1,2,3]. In order to solve this problem, energy harvesting technology is emerging as the times require [4,5,6]. Energy harvesting is the key to realizing green and sustainable communication, which can solve the contradiction between unlimited energy supply and limited IoT battery capacity and prolong the life of IoT devices [7,8]. However, the power supply or data transmission by the base station to the device is easily blocked by obstacles, which reduces the charging efficiency and transmission quality. To solve this problem, Reconfigurable Intelligent Surface (RIS) was proposed as a new technology [9,10,11,12]. Specifically, RIS consists of a large number of low-cost passive reflective elements, which can independently adjust the phase shift and amplitude of the incident signal, thereby changing the reflection route [13]. Because RIS is easy to deploy and can be reconfigured through passive beamforming, integrating it into existing communication systems is an effective way to solve the above problems.

Therefore, in recent years, a large number of scholars have conducted in-depth research on RIS-assisted communication technology. For example, ref. [14] considers the service quality constraints of the nonlinear energy harvesting receiver and the information decoding receiver, and jointly optimizes the base station beamforming vector and the transmission mode selection of each region of the RIS to minimize the transmit power of the base station. Reference [15] considers the maximum transmit power constraint, the minimum energy harvesting constraint and the RIS phase shift constraint of the base station and proposes a multi-objective optimization framework, which maximizes the system sum rate by jointly optimizing the base station energy, information beamforming and RIS phase shift. However, the above work does not consider the harvested energy efficiency optimization problem. Based on this, ref. [16] considers the minimum rate constraint, energy harvesting constraint and transmission power constraint and jointly optimizes the information and energy beamforming of the base station, RIS phase shift and power allocation ratio, the maximum minimum system energy efficiency. For the RIS-assisted wireless power supply communication system, based on the bounded channel uncertainty, ref. [17] maximizes the robust energy efficiency of the system by jointly optimizing the energy beam and RIS phase shift. However, the above work does not consider the issue of network information security. Based on this, ref. [18] studied the RIS-assisted multi-user Multiple-Input Single-Output (MISO) secure communication problem considering artificial noise, which aims to maximize the total secrecy rate of the system by jointly optimizing the RIS phase shift, the beamforming vector at the base station and the artificial noise. This work ensures security, but cannot be directly applied to wireless energy-carrying communication systems. For the RIS-assisted MISO system, ref. [19] considered the impact of transceiver hardware impairment and developed a novel secure energy efficiency maximization algorithm, while ensuring the constraints of the base station transmit power and the RIS modulo-one constraint. In [20], a multi-antenna full-duplex jammer is considered for the RIS-assisted single-input multiple-output communication system, which exploits the block coordinate descent method to study the problem of maximizing secure energy efficiency. Reference [21] studies a number-capable simultaneous interpretation algorithm for the RIS-assisted MISO SWIPT network, which maximizes the secure rate and speed of network weighting. However, the above work assumes that the Channel State Information (CSI) is perfectly known and ignores the influence of channel uncertainty. Due to the influence of channel estimation error, quantization error and feedback delay, it is difficult for the base station to obtain accurate CSI [22], so that the designed algorithm often does not meet the actual needs in the actual system.

In order to improve the harvested energy of the system and overcome the influence of channel uncertainty and security risks caused by eavesdroppers, this paper proposes a robust secure resource allocation algorithm for RIS-assisted MISO SWIPT systems with channel uncertainties in the presence of multiple single-antenna potential eavesdroppers. The main contributions are as follows:For an RIS-assisted MISO communication system consisting of multiple energy receiving devices, multiple information receivers and multiple eavesdroppers, a joint optimization of multi-variable coupling of information beam, energy beam, RIS phase shift, nonlinear energy efficiency maximization resource allocation problem is investigated considering the security rate constraint, the maximum transmit power constraint and the minimum energy harvesting constraint. This problem is a nonlinear and non-convex optimization problem with parameter perturbation and it is not easy to obtain an analytical solution.In order to solve this problem, the S-Procedure method is used to convert channel uncertainty constraints into deterministic constraints; on this basis, alternating optimization methods are used to convert non-convex problems into deterministic and convex optimization problems and propose an alternating optimization algorithm based on continuous convex approximation.Simulation results show that the proposed algorithm has good convergence, energy efficiency and robustness.

## 2. Network Model and Problem Formulation

A multi-user RIS-assisted MISO communication system is considered in this paper, which consists of a base station with *M* antennas, an RIS with *N* reflection units, *K* single-antenna legitimate users, *L* eavesdroppers and *R* energy harvesting devices. Every user device or eavesdropper receives the base, the direct signal from the station and the reflected signal from the RIS. Define the set of legitimate users as K={1,2,…,K}(∀k∈K), the set of eavesdroppers as L={1,2,…,L}(∀l∈L), the set of energy harversting devices as R={1,2,…,R}(∀r∈R) and the set of reflection elements as N={1,2,…,N}(∀n∈N). We define $H^{BI}\in C^{N\times M}$ as the channel between base station and RIS. $h_{k}^{BU}\in C^{M\times 1}$, $h_{l}^{BE}\in C^{M\times M}$ and $g_{r}^{BH}\in C^{M\times 1}$ denote the channel between base station and *k*-th legal user, *l*-th eavesdropper and *r*-th energy harvesting device, respectively, while $h_{k}^{IU}\in C^{N\times 1}$, $h_{l}^{IE}\in C^{N\times 1}$ and $g_{r}^{IH}\in C^{N\times 1}$ represent the channel between RIS and *k*-th legal user, *l*-th eavesdropper and *r*-th energy harvesting device. Θ=diag(q) is defined as the phase shift matrix, where $q=[q_{1},q_{2},\ldots ,q_{N}]$, $q_{n}=e^{j\theta_{n}}$, $\theta_{n}\in \left[0,2\pi \right]$.

It is assumed that each information receiver and energy harvesting device is equipped with unique information and energy beam; then, the received signal at the base station can be written as
(1)$$x=\sum_{k=1}^{K}w_{k}s_{k}^{ID}+\sum_{r=1}^{R}v_{r}s_{r}^{EH}$$
where $s_{k}^{ID}$ and $s_{r}^{EH}$ represent the information signal sent by the base station to the *k*-th legal user and the energy signal of the *r*-th energy harvesting device, satisfying $E\{|s_{k}^{ID}{|}^{2}\}=1$ and $E\{|s_{r}^{EH}{|}^{2}\}=1$. The vector $w_{k}\in C^{M\times 1}$ and $v_{r}\in C^{M\times 1}$ denote the information beam and energy beam from the base station to the *k*-th legal user, respectively. Therefore, the total transmit power of the base station can be represented as
(2)$$E\left(x^{H}x\right)=\sum_{k=1}^{K}∥w_{k}{∥}^{2}+\sum_{r=1}^{R}{∥v_{r}∥}^{2}$$

According to the system model diagram and the above signal definition, the received signal of the *k*-th legitimate user and the corresponding eavesdropping signal of the *l*-th eavesdropper can be expressed as follows:(3)$$y_{k}=\left({\left(h_{k}^{\mathrm{IU}}\right)}^{H}\Theta H^{\mathrm{BI}}+{\left(h_{k}^{\mathrm{BU}}\right)}^{H}\right)x+n_{k}$$
(4)$$y_{e,l}=\left({\left(h_{l}^{\mathrm{IE}}\right)}^{H}\Theta H^{\mathrm{BI}}+{\left(h_{l}^{\mathrm{BE}}\right)}^{H}\right)x+n_{l}$$
where $n_{k}\;\mathrm{CN}\left(0,\sigma_{k}^{2}\right)$ and $n_{l}\;\mathrm{CN}\left(0,\delta_{l}^{2}\right)$ denote the noise of the *k*-th legitimate user and *l*-th eavesdropper, respectively. Then, the rate of the *k*-th legitimate user and *l*-th eavesdropper can be expressed as
(5)$$R_{k}=B\log_{2}\left(1+\gamma_{k}\right)$$
(6)$$R_{k,l}=B\log_{2}\left(1+\gamma_{k,l}\right)$$
where $\gamma_{k}=\frac{|h_{k}^{H}w_{k}{|}^{2}}{\sum_{j\neq k}^{K}|h_{k}^{H}w_{j}{|}^{2}+\sum_{r=1}^{R}{\left|h_{k}^{H}v_{r}\right|}^{2}+\delta_{k}^{2}}$, $\gamma_{k,l}=\frac{|g_{l}^{H}w_{k}{|}^{2}}{\sum_{j\neq k}^{K}|g_{l}^{H}w_{j}{|}^{2}+\sum_{r=1}^{R}{\left|g_{l}^{H}v_{r}\right|}^{2}+\sigma_{k}^{2}}$, $h_{k}^{H}={\left(h_{k}^{IU}\right)}^{H}\Theta H^{\mathrm{BI}}$ + ${\left(h_{k}^{\mathrm{BU}}\right)}^{H}$, $g_{l}^{H}={\left(h_{l}^{IE}\right)}^{H}\Theta H^{\mathrm{BI}}+{\left(h_{l}^{\mathrm{BE}}\right)}^{H}$. For security provisioning in our considered systems, we adopt the worst-case assumption on the eavesdropping capabilities of potential eavesdroppers. Therefore, the channel capacity between the AP and the *l*-th potential eavesdropper for decoding the signal of the *k*-th legitimate user is given by
(7)$$R_{k}^{\sec}={\left(R_{k}-\max_{\forall l\in L}R_{k,l}\right)}^{+}$$
where ${\left(x\right)}^{+}=\max\left(0,x\right)$. Due to the low power of background noise power, we ignore the influence of noise power. Hence, the energy collected by each energy harvesting device is
(8)$$E_{r}=\rho \left(\sum_{k=1}^{K}∥G_{r}^{H}w_{k}{∥}^{2}+\sum_{r=1}^{R}{∥G_{r}^{H}v_{r}∥}^{2}\right)$$
where ρ∈[0,1] is the energy conversion efficiency factor. Without loss of generality, ρ is set to 1 in the paper.

Since the eavesdroppers are randomly distributed in the network and do not have any cooperation with the base station, the relevant CSI cannot be estimated accurately. Here, we consider the following bounded eavesdropping channel uncertainty model:(9)$$\begin{array}{cc} \; & h_{l}^{\mathrm{BE}}={\bar{h}}_{l}^{\mathrm{BE}}+\Delta h_{l}^{\mathrm{BE}},h_{l}^{\mathrm{IE}}={\bar{h}}_{l}^{\mathrm{IE}}+\Delta h_{l}^{\mathrm{IE}},\;\\ \; & \Omega_{E}=\{∥\Delta h_{l}^{\mathrm{BE}}{∥}_{F}\leq \varepsilon_{l}^{\mathrm{BE}},∥\Delta h_{l}^{\mathrm{IE}}{∥}_{F}\leq \varepsilon_{l}^{\mathrm{IE}}\} \end{array}$$
where ${\bar{h}}_{l}^{\mathrm{BE}}\in C^{M\times 1}$ and ${\bar{h}}_{l}^{\mathrm{IE}}\in C^{N\times 1}$ denote the estimated channel, $\Delta h_{l}^{\mathrm{BE}}\in C^{M\times 1}$ and $\Delta h_{l}^{\mathrm{IE}}\in C^{N\times 1}$ are the corresponding channel uncertainties, $\varepsilon_{l}^{\mathrm{BE}}$ and $\varepsilon_{l}^{\mathrm{IE}}$ represent the corresponding upper bound of the estimated error. Based on the above description, the robust resource allocation problem of maximizing fairness-aware harvested energy of the system can be described as
(10)$$\begin{array}{cc} \; & \max_{w_{k},v_{r},\theta_{n}}\min_{\forall r\in R}\left(\sum_{k=1}^{K}∥G_{r}^{H}w_{k}{∥}^{2}+\sum_{r=1}^{R}{∥G_{r}^{H}v_{r}∥}^{2}\right)\\ \; & s.t.\;C1:\;R_{k}^{\sec}\geq R_{k}^{\min},∥\Delta h_{l}^{\mathrm{BE}}{∥}_{F}\leq \varepsilon_{l}^{\mathrm{BE}},{∥\Delta h_{l}^{\mathrm{IE}}∥}_{F}\leq \varepsilon_{l}^{\mathrm{IE}}\\ \; & \;\;\;\;\;\;C2:\;\sum_{k=1}^{K}∥w_{k}{∥}^{2}+\sum_{r=1}^{R}{∥v_{r}∥}^{2}\leq P_{\max}\\ \; & \;\;\;\;\;\;C3:\;0\leq \theta_{n}\leq 2\pi \end{array}$$
where C1 represents the minimum security rate constraint of the legal user *k* and $R_{k}^{\min}$ is the corresponding minimum security rate threshold; C2 represents the maximum transmit power constraint of the base station, where $P_{\max}$ is the maximum transmit power of the base station; C3 represents the phase shift constraint of RIS. Since the objective function in Equation (Equation 10) is non-concave form and there are coupled variables in constraints C1 and C2, Equation (Equation 10) is a non-convex optimization problem.

## 3. Robust Resource Allocation Algorithm

Obviously, the formulated optimization problem (Equation 10) cannot be directly solved by exploiting the standard convex theory. In order to obtain the solution of the considered problem (Equation 10), we firstly utilize the the S-procedure method to transform the channel uncertainty constraints into deterministic linear constraints. Then, the solution is obtained using an alternating optimization method. In order to remove the coupling between optimization variables and make the problem easy to solve, the active beamformer at the base station is optimized as the passive phase shift at the RIS is fixed, then the passive phase shift is optimized as the active beamformer at the base station is fixed.

### 3.1. Optimizing Active Beamformer

As the Θ is fixed, the channels between base station and the legal users as well as the the channels from the base station to the energy harvesting users are fixed. For further processing, we define $W_{k}=w_{k}w_{k}^{H}$, $H_{k}=h_{k}h_{k}^{H}$, $V_{r}=v_{r}v_{r}^{H}$, ${\bar{G}}_{r}=G_{r}G_{r}^{H}$. Then, the optimization problem (Equation 10) subject to $W_{k}$ and $V_{r}$ can be reformulated as
(11)$$\begin{array}{cc} \; & \max_{W_{k},V_{r}}\min_{\forall r\in R}\;\;\left(\sum_{k=1}^{K}\mathrm{Tr}\left({\bar{G}}_{r}W_{k}\right)+\sum_{m=1}^{R}\mathrm{Tr}\left({\bar{G}}_{r}V_{m}\right)\right)\\ \; & s.t.\;C1:\;R_{k}-\max_{\forall l\in L}R_{k,l}\geq R_{k}^{\min},∥\Delta h_{l}^{\mathrm{BE}}{∥}_{F}\leq \varepsilon_{l}^{\mathrm{BE}},{∥\Delta h_{l}^{\mathrm{IE}}∥}_{F}\leq \varepsilon_{l}^{\mathrm{IE}}\\ \; & \;\;\;\;\;\;\bar{C2}:\sum_{k=1}^{K}\mathrm{Tr}\left(W_{k}\right)+\sum_{r=1}^{R}\mathrm{Tr}\left(V_{r}\right)\leq P_{\max}\\ \; & \;\;\;\;\;\;C4:W_{k}⪰0,V_{r}⪰0\\ \; & \;\;\;\;\;\;C5:\mathrm{Rank}(W_{k})=1 \end{array}$$

The slack variable *t* is introduced; then, the above optimization problem can be expressed as
(12)$$\begin{array}{cc} \; & \max_{W_{k},V_{r}}\;\;t\\ \; & s.t.\;C1:\;R_{k}-\max_{\forall l\in L}R_{k,l}\geq R_{k}^{\min},∥\Delta h_{l}^{\mathrm{BE}}{∥}_{F}\leq \varepsilon_{l}^{\mathrm{BE}},{∥\Delta h_{l}^{\mathrm{IE}}∥}_{F}\leq \varepsilon_{l}^{\mathrm{IE}}\\ \; & \;\;\;\;\;\;\bar{C2}:\sum_{k=1}^{K}\mathrm{Tr}\left(W_{k}\right)+\sum_{r=1}^{R}\mathrm{Tr}\left(V_{r}\right)\leq P_{\max}\\ \; & \;\;\;\;\;\;C4:W_{k}⪰0,V_{r}⪰0\\ \; & \;\;\;\;\;\;C5:\mathrm{Rank}(W_{k})=1\\ \; & \;\;\;\;\;\;C6:\left(\sum_{k=1}^{K}\mathrm{Tr}\left({\bar{G}}_{r}W_{k}\right)+\sum_{m=1}^{R}\mathrm{Tr}\left({\bar{G}}_{r}V_{m}\right)\right)>=t \end{array}$$

Furthermore, another slack variable $\tau_{k,l}^{\max}$ is introduced; then, the constraint C2 can be expressed as
(13)$$\begin{array}{cc} \; & \bar{C1}:R_{k}-\tau_{k,l}^{\max}\geq R_{k}^{\min},R_{k,l}\leq \tau_{k,l}^{\max}\\ \; & ∥\Delta h_{l}^{\mathrm{BE}}{∥}_{F}\leq \varepsilon_{l}^{\mathrm{BE}},{∥\Delta h_{l}^{\mathrm{IE}}∥}_{F}\leq \varepsilon_{l}^{\mathrm{IE}} \end{array}$$
where $\tau_{k,l}^{\max}$ is the maximum eavesdrop rate of each eavesdropper. The eavesdrop rate is represented as
(14)$$R_{k,l}=B\log_{2}\left(\frac{\sum_{k=1}^{K}|g_{l}^{H}w_{k}{|}^{2}+\sum_{r=1}^{R}{\left|g_{l}^{H}v_{r}\right|}^{2}+\sigma_{l}^{2}}{\sum_{j\neq k}^{K}|g_{l}^{H}w_{j}{|}^{2}+\sum_{r=1}^{R}{\left|g_{l}^{H}v_{r}\right|}^{2}+\sigma_{l}^{2}}\right)<=\tau_{k,l}^{\max}$$
The legal information rate $R_{k}$ can be formulated as
(15)$$\begin{array}{cc} R_{k} & =B\log_{2}\left(\sum_{k=1}^{K}\mathrm{Tr}\left(H_{k}W_{k}\right)+\sum_{r=1}^{R}\mathrm{Tr}\left(H_{k}V_{r}\right)+\delta_{k}^{2}\right)\\ \; & -B\log_{2}\left(\sum_{j\neq k}^{K}\mathrm{Tr}\left(H_{k}W_{j}\right)+\sum_{r=1}^{R}\mathrm{Tr}\left(H_{k}V_{r}\right)+\delta_{k}^{2}\right) \end{array}$$
However, the optimization problem (Equation 12) belongs to the DC programming problem, the second term in (Equation 15) is processed using the first-order Taylor expansion. Defining the second term in (Equation 15) as $D_{1}\left(W_{k},V_{r}\right)$, we have
(16)$$\begin{array}{cc} \; & D_{1}\left(W_{k},V_{r}\right)\geq D_{1}\left(W_{k}^{\left(i\right)},V_{r}^{\left(i\right)}\right)\\ \; & +\mathrm{Tr}(\nabla_{V_{r}^{\left(i\right)}}^{H}D_{1}\left(W_{k}^{\left(i\right)},V_{r}^{\left(i\right)}\right)\left(V_{r}-V_{r}^{\left(i\right)}\right))\\ \; & +\sum_{k=1}^{K}\mathrm{Tr}\left(\nabla_{W_{k}^{\left(i\right)}}^{H}D_{1}\left(W_{k}^{\left(i\right)},V_{r}^{\left(i\right)}\right)\left(W_{k}-W_{k}^{i}\right)\right) \end{array}$$
where
(17)$$\begin{array}{cc} \; & \nabla_{V_{r}^{\left(i\right)}}^{H}D_{1}\left(W_{k}^{\left(i\right)},V_{r}^{\left(i\right)}\right)\\ \; & =\frac{B}{\ln2}\sum_{k=1}^{K}\frac{H_{k}}{\sum_{j\neq k}^{K}\mathrm{Tr}\left(H_{k}W_{j}\right)+\sum_{r=1}^{R}\mathrm{Tr}\left(H_{k}V_{r}\right)+\delta_{k}^{2}} \end{array}$$
(18)$$\begin{array}{cc} \; & \nabla_{W_{k}^{\left(i\right)}}^{H}D_{1}\left(W_{k}^{\left(i\right)},V_{r}^{\left(i\right)}\right)\\ \; & =\frac{B}{\ln2}\sum_{k\neq j}^{K}\frac{H_{k}}{\sum_{j\neq k}^{K}\mathrm{Tr}\left(H_{k}W_{j}\right)+\sum_{r=1}^{R}\mathrm{Tr}\left(H_{k}V_{r}\right)+\delta_{k}^{2}} \end{array}$$

Since the objective function of Equation (Equation 12) is maximization, the lower bound of $D_{1}\left(W_{k},V_{r}\right)$ is taken to replace it; then, $R_{k}$ can be expressed as
(19)$$\begin{array}{cc} {\hat{R}}_{k} & =B\log_{2}\left(\sum_{j=1}^{K}\mathrm{Tr}\left(H_{k}W_{j}\right)+\sum_{r=1}^{R}\mathrm{Tr}\left(H_{k}V_{r}\right)+\delta_{k}^{2}\right)\\ \; & -\mathrm{Tr}\left(\nabla_{V_{r}^{\left(i\right)}}^{H}D_{1}\left(W_{k}^{\left(i\right)},V_{r}^{\left(i\right)}\right)\left(V_{r}-V_{r}^{\left(i\right)}\right)\right)\\ \; & -\sum_{k=1}^{K}\mathrm{Tr}\left(\nabla_{W_{k}^{\left(i\right)}}^{H}D_{1}\left(W_{k}^{\left(i\right)},V_{r}^{\left(i\right)}\right)\left(W_{k}-W_{k}^{i}\right)\right) \end{array}$$
However, (Equation 12) contains uncertainty constraints, which are difficult to handle. To make the problem easy to solve, the new optimization variable β is introduced, which is defined as
(20)$$\begin{array}{c} \left[\begin{array}{cccc} \beta_{1,1} & \beta_{1,2} & \ldots & \beta_{1,L}\\ \beta_{2,1} & \beta_{2,2} & \ldots & \beta_{2,L}\\ \vdots & \vdots & \ddots & \vdots \\ \beta_{K,1} & \beta_{K,2} & \ldots & \beta_{K,L} \end{array}\right] \end{array}$$
Then, the eavesdrop rate $R_{k,l}$ of the *k*-th eavesdropper can be equivalently transformed into the following expression:(21)$$\begin{array}{cc} \; & \sum_{k=1}^{K}|g_{l}^{H}w_{k}{|}^{2}+\sum_{r=1}^{R}{\left|g_{l}^{H}v_{r}\right|}^{2}+\delta_{l}^{2}\leq \beta_{k,l}2^{\frac{\tau_{k,l}^{\max}}{B}},\\ \; & \Delta h_{l}^{\mathrm{BE}},\Delta h_{l}^{\mathrm{IE}}\in \Omega_{E} \end{array}$$
(22)$$\begin{array}{cc} \; & \sum_{j\neq k}^{K}|g_{l}^{H}w_{j}{|}^{2}+\sum_{r=1}^{R}{\left|g_{l}^{H}v_{r}\right|}^{2}+\delta_{l}^{2}\geq \beta_{k,l},\\ \; & \Delta h_{l}^{\mathrm{BE}},\Delta h_{l}^{\mathrm{IE}}\in \Omega_{E} \end{array}$$
Moreover, (Equation 21) and (Equation 22) can be rewritten as
(23)(h¯lIE)HΘHBI+h¯lBEH∑k=1KWk+∑r=1RVr(h¯lIE)HΘHBI+h¯lBEHH+2Re{(h¯lIE)HΘHBI+h¯lBEH∑k=1KWk+∑r=1RVr(ΔhlBE+ΔhlIE)}+(ΔhlBE+ΔhlIE)H∑k=1KWk+∑r=1RVr+(ΔhlBE+ΔhlIE)H∑k=1KWk+∑r=1RVr(ΔhlBE+ΔhlIE)+δl2≤βk,l2τk,lmaxB
(24)(h¯lIE)HΘHBI+h¯lBEH∑j≠kKWj+∑r=1RVr(h¯lIE)HΘHBI+h¯lBEHH+2Re{(h¯lIE)HΘHBI+h¯lBEH∑j≠kKWj+∑r=1RVr(ΔhlBE+ΔhlIE)}+(ΔhlBE+ΔhlIE)H∑j≠kKWj+∑r=1RVr+(ΔhlBE+ΔhlIE)H∑j≠kKWj+∑r=1RVr(ΔhlBE+ΔhlIE)+δl2≤βk,l2τk,lmaxB

The estimated channel vector and channel estimation error of the eavesdropper are defined as
(25)$${\bar{g}}_{l}^{H}=\left[{\left({\bar{h}}_{l}^{\mathrm{IE}}\right)}^{H},{\left({\bar{h}}_{l}^{\mathrm{BE}}\right)}^{H}\right],\Delta g_{l}^{H}=\left[{\left(\Delta h_{l}^{\mathrm{IE}}\right)}^{H},{\left(\Delta h_{l}^{\mathrm{IE}}\right)}^{H}\right]$$
Hence, (Equation 23) and (Equation 24) can be rewritten as
(26)ΔglHΓΔgl+2Re{ΔglHΓg¯l}+g¯lHΓg¯l+δl2−βk,lτk,lmaxB≤0,∥Δgl∥F≤εlE
(27)ΔglHΓkΔgl+2Re{ΔglHΓkg¯l}+g¯lHΓkg¯l+δl2−βk,lτk,lmaxB≤0,∥Δgl∥F≤εlE
where $\varepsilon_{l}^{E}=\varepsilon_{l}^{\mathrm{BE}}+\varepsilon_{l}^{\mathrm{IE}}$. Since Equations (Equation 26) and (Equation 27) contain infinite linear inequalities, in order to deal with this problem, Theorem 1 is used for transformation.

**Theorem** **1**(S-Procedure [16])**.**
*Define $f_{1}\left(x\right)$ and $f_{2}\left(x\right)$ as*
(28)fm(x)=xHAmx+2Re{bmHx}+cm,m=1,2
*where $A_{m}\in R^{N\times N}$, $b_{m}\in R^{N\times 1}$, $x\in R^{N\times 1}$, $c_{m}\in R$. If and only if λ≥0, $f_{1}\left(x\right)\leq 0\Rightarrow f_{2}\left(x\right)\leq 0$, which yields*
(29)$$\begin{array}{c} \lambda \left[\begin{array}{cc} A_{1} & b_{1}\\ b_{1}^{H} & c_{1} \end{array}\right]-\left[\begin{array}{cc} A_{2} & b_{2}\\ b_{2}^{H} & c_{2} \end{array}\right]⪰0, \end{array}$$

Applying Theorem 1, (Equation 26) and (Equation 27) convert to a finite number of linear matrix inequalities, such as
(30)$$\begin{array}{c} \left[\begin{array}{cc} \lambda_{l}I_{N+M}-\Gamma & -\Gamma^{H}{\bar{g}}_{l}\\ -{\left(\Gamma^{H}{\bar{g}}_{l}\right)}^{H} & -\lambda_{l}{\left(\varepsilon_{l}^{E}\right)}^{2}-{\bar{g}}_{l}^{H}\Gamma {\bar{g}}_{l}-Z_{1} \end{array}\right]⪰0. \end{array}$$
(31)$$\begin{array}{c} \left[\begin{array}{cc} {\bar{\lambda }}_{l}I_{N+M}-\Gamma_{k} & -\Gamma_{k}^{H}{\bar{g}}_{l}\\ -{\left(\Gamma_{k}^{H}{\bar{g}}_{l}\right)}^{H} & -{\bar{\lambda }}_{l}{\left(\varepsilon_{l}^{E}\right)}^{2}-{\bar{g}}_{l}^{H}\Gamma_{k}{\bar{g}}_{l}-Z_{1} \end{array}\right]⪰0. \end{array}$$
where
(32)$$\begin{array}{c} \Gamma =\left[\begin{array}{cc} \Theta H^{\mathrm{BI}}\left(\sum_{j=1}^{K}W_{j}+\sum_{r=1}^{R}V_{r}\right){\left(H^{\mathrm{BI}}\right)}^{H}G & \Theta H^{\mathrm{BI}}\left(\sum_{j=1}^{K}W_{j}+\sum_{r=1}^{R}V_{r}\right)\\ {\left(\Theta H^{\mathrm{BI}}\left(\sum_{j=1}^{K}W_{j}+\sum_{r=1}^{R}V_{r}\right)\right)}^{H} & \left(\sum_{j=1}^{K}W_{j}+\sum_{r=1}^{R}V_{r}\right) \end{array}\right]⪰0. \end{array}$$
(33)$$\begin{array}{c} \Gamma_{k}=\left[\begin{array}{cc} \Theta H^{\mathrm{BI}}\left(\sum_{j\neq k}^{K}W_{j}+\sum_{r=1}^{R}V_{r}\right){\left(H^{\mathrm{BI}}\right)}^{H}G & \Theta H^{\mathrm{BI}}\left(\sum_{j\neq k}^{K}W_{j}+\sum_{r=1}^{R}V_{r}\right)\\ {\left(\Theta H^{\mathrm{BI}}\left(\sum_{j\neq k}^{K}W_{j}+\sum_{r=1}^{R}V_{r}\right)\right)}^{H} & \left(\sum_{j\neq k}^{K}W_{j}+\sum_{r=1}^{R}V_{r}\right) \end{array}\right]⪰0. \end{array}$$
$Z_{1}=\delta_{l}^{2}-\beta_{k,l}2^{\frac{\tau_{k,l}^{\max}}{B}}$, $Z_{2}=\delta_{l}^{2}-\beta_{k,l}$, $\lambda_{l}$ and ${\bar{\lambda }}_{l}$ denote the new slack variables. Let $\lambda =[\lambda_{1},\lambda_{2},\ldots ,\lambda_{L}]$ and $\bar{\lambda }=\left[{\bar{\lambda }}_{1},{\bar{\lambda }}_{2},\ldots ,{\bar{\lambda }}_{L}\right]$. Then, the optimization problem (12) can be reformulated as
(34)$$\begin{array}{cc} \; & \max_{W_{k},V_{r},t,\beta_{k,l},\lambda_{l},\bar{\lambda }}\;\;\;t\\ \; & s.t.:\;\;\bar{C}2,C4,C5,C6\\ \; & \;\;\;\;\;\;{\hat{R}}_{k}-\tau_{k,l}^{\max}>=R_{k}^{\min}\\ \; & \;\;\;\left[\begin{array}{cc} \lambda_{l}I_{N+M}-\Gamma & -\Gamma^{H}{\bar{g}}_{l}\\ -{\left(\Gamma^{H}{\bar{g}}_{l}\right)}^{H} & -\lambda_{l}{\left(\varepsilon_{l}^{E}\right)}^{2}-{\bar{g}}_{l}^{H}\Gamma {\bar{g}}_{l}-Z_{1} \end{array}\right]⪰0\\ \; & \;\;\;\left[\begin{array}{cc} {\bar{\lambda }}_{l}I_{N+M}-\Gamma_{k} & -\Gamma_{k}^{H}{\bar{g}}_{l}\\ -{\left(\Gamma_{k}^{H}{\bar{g}}_{l}\right)}^{H} & -{\bar{\lambda }}_{l}{\left(\varepsilon_{l}^{E}\right)}^{2}-{\bar{g}}_{l}^{H}\Gamma_{k}{\bar{g}}_{l}-Z_{1} \end{array}\right]⪰0 \end{array}$$
It can be easily known that (Equation 34) is still a convex problem only due to the non-convex constraint $\mathrm{Rank}(W_{k})=1$. Herein, we adopt the semidefinite relaxation technique (SDR), dropping the constraint $\mathrm{Rank}(W_{k})=1$. Therefore, the above problem is a convex problem, which can be solved by the standard convex solver, such as, CVX. However, this does not guarantee $\mathrm{Rank}(W_{k})=1$. If $\mathrm{Rank}(W_{k})=1$, the optimal solution $w_{k}$ can be obtained by eigenvalue decomposition. Otherwise, a rank-one solution can be constructed by the Gaussian randomization method.

### 3.2. Optimizing Phase Shift

As $w_{k}$, $v_{r}$ and t are fixed, the optimization problem (12) subject to phase shift $\theta_{n}$ can be reformulated as
(35)$$\begin{array}{cc} \; & \max_{t,\theta_{n}}\;\;t\\ \; & s.t.\;C1:\;R_{k}^{\sec}\geq R_{k}^{\min},∥\Delta h_{l}^{\mathrm{BE}}{∥}_{F}\leq \varepsilon_{l}^{\mathrm{BE}},{∥\Delta h_{l}^{\mathrm{IE}}∥}_{F}\leq \varepsilon_{l}^{\mathrm{IE}}\\ \; & \;\;\;\;\;\;\left(\sum_{k=1}^{K}∥G_{r}^{H}w_{k}{∥}^{2}+\sum_{r=1}^{R}{∥G_{r}^{H}v_{r}∥}^{2}\right)>=t\\ \; & \;\;\;\;\;\;C3:\;0\leq \theta_{n}\leq 2\pi \end{array}$$

Define $\bar{q}=\left[q;1\right]$, $Q=\bar{q}{\bar{q}}^{H}$, $\Phi_{k}=\left[\mathrm{diag}\left({\left(h_{k}^{\mathrm{IU}}\right)}^{H}\right)H^{\mathrm{BI}};{\left(h_{k}^{\mathrm{BU}}\right)}^{H}\right]$, $\Phi_{l}=[\mathrm{diag}\left({\left(h_{l}^{\mathrm{IE}}\right)}^{H}\right)H^{\mathrm{BI}}$; ${\left(h_{l}^{\mathrm{BE}}\right)}^{H}]$, $\Phi_{r}=\left[\mathrm{diag}\left({\left(g_{r}^{\mathrm{IH}}\right)}^{H}\right)H^{\mathrm{BI}};{\left(g_{r}^{\mathrm{BH}}\right)}^{H}\right]$; then, (Equation 35) becomes
(36)$$\begin{array}{cc} \; & \max_{t,\lambda_{l},\bar{\lambda },Q}\;\;t\\ \; & s.t.\;\;{\tilde{R}}_{k}-\tau_{k,l}^{\max}\geq R_{k}^{\min},\\ \; & \;\;\;\;\;\;\left(\sum_{k=1}^{K}\mathrm{Tr}\left(\Phi_{r}W_{k}{\left(\Phi_{r}\right)}^{H}Q\right)+\sum_{m=1}^{R}\mathrm{Tr}\left(\Phi_{r}V_{m}{\left(\Phi_{r}\right)}^{H}Q\right)\right)>=t\\ \; & \;\;\;\;\;\;\hat{C3}:\;Q_{n,n}=1,Q⪰0\\ \; & \;\;\;\left[\begin{array}{cc} \lambda_{l}I_{N+M}-\Gamma & -\Gamma^{H}{\bar{g}}_{l}\\ -{\left(\Gamma^{H}{\bar{g}}_{l}\right)}^{H} & -\lambda_{l}{\left(\varepsilon_{l}^{E}\right)}^{2}-{\bar{g}}_{l}^{H}\Gamma {\bar{g}}_{l}-Z_{1} \end{array}\right]⪰0\\ \; & \;\;\;\left[\begin{array}{cc} {\bar{\lambda }}_{l}I_{N+M}-\Gamma_{k} & -\Gamma_{k}^{H}{\bar{g}}_{l}\\ -{\left(\Gamma_{k}^{H}{\bar{g}}_{l}\right)}^{H} & -{\bar{\lambda }}_{l}{\left(\varepsilon_{l}^{E}\right)}^{2}-{\bar{g}}_{l}^{H}\Gamma_{k}{\bar{g}}_{l}-Z_{1} \end{array}\right]⪰0\\ \; & \;\;\;\mathrm{Rank}(Q)=1 \end{array}$$
where
(37)$$\begin{array}{cc} {\tilde{R}}_{k} & =B\log_{2}\left(\sum_{k=1}^{K}\mathrm{Tr}\left(\Phi_{k}W_{k}{\left(\Phi_{k}\right)}^{H}Q\right)+\sum_{r=1}^{R}\mathrm{Tr}\left(\Phi_{k}V_{r}{\left(\Phi_{k}\right)}^{H}Q\right)+\sigma_{k}^{2}\right)\\ \; & -\mathrm{Tr}\left({▽}_{Q}^{H}{\tilde{D}}_{1}\left(Q^{\left(i\right)}\right)\left(Q-Q^{\left(i\right)}\right)\right) \end{array}$$
(38)$$\begin{array}{cc} \; & {▽}_{Q}^{H}{\tilde{D}}_{1}\left(Q^{\left(i\right)}\right)=\\ \; & \frac{B}{\ln2}\sum_{k=1}^{K}\frac{\Phi_{k}\left(\sum_{j\neq k}^{K}W_{j}+\sum_{r=1}^{R}V_{r}\right){\left(\Phi_{k}\right)}^{H}}{\left(\sum_{j\neq k}^{K}\mathrm{Tr}\left(\Phi_{k}W_{j}{\left(\Phi_{k}\right)}^{H}Q\right)+\sum_{r=1}^{R}\mathrm{Tr}\left(\Phi_{k}V_{r}{\left(\Phi_{k}\right)}^{H}Q\right)+\sigma_{k}^{2}\right)} \end{array}$$
For the non-convex terms Γ and $\Gamma_{k}$ in (Equation 36), we use singular value decomposition to transform, for example,
(39)$$H^{\mathrm{BI}}\left(\sum_{j=1}^{K}W_{j}+\sum_{r=1}^{R}V_{r}\right){\left(H^{\mathrm{BI}}\right)}^{H}=\sum_{i}s_{i}d_{i}^{H}$$
(40)$$H^{\mathrm{BI}}\left(\sum_{j\neq k}^{K}W_{j}+\sum_{r=1}^{R}V_{r}\right){\left(H^{\mathrm{BI}}\right)}^{H}=\sum_{i}{\bar{s}}_{i}{\bar{d}}_{i}^{H}$$

Moreover, we have
(41)$$\Theta^{H}H^{\mathrm{BI}}\left(\sum_{j=1}^{K}W_{j}+\sum_{r=1}^{R}V_{r}\right){\left(H^{\mathrm{BI}}\right)}^{H}\Theta =\sum_{i}S_{i}\bar{q}{\left(\bar{q}\right)}^{H}D_{i}^{H}$$
(42)$$\Theta^{H}H^{\mathrm{BI}}\left(\sum_{j\neq k}^{K}W_{j}+\sum_{r=1}^{R}V_{r}\right){\left(H^{\mathrm{BI}}\right)}^{H}\Theta =\sum_{i}{\bar{S}}_{i}\bar{q}{\left(\bar{q}\right)}^{H}{\bar{D}}_{i}^{H}$$
where $S_{i}=\left[\mathrm{diag}{\left(s\right)}_{i},0\right]$, ${\bar{S}}_{i}=\left[\mathrm{diag}\left({\bar{s}}_{i}\right),0\right]$, $D_{i}=\left[\mathrm{diag}\left(d_{i}\right);0\right]$,${\bar{D}}_{i}=\left[\mathrm{diag}\left({\bar{d}}_{i}\right);0\right]$. If Q satisfies Rank(Q)=1, then $\Theta =\mathrm{diag}(Q_{N+1,1:N})$, where $Q_{N+1,1:N}=\left[Q_{N+1,1},Q_{N+1,2},\ldots ,Q_{N+1,N}\right]$. Based on the above analysis, Γ and $\Gamma_{k}$ in (Equation 34) can be re-expressed as
(43)$$\begin{array}{cc} \hat{\Gamma }= & \left[\begin{array}{cc} \sum_{i}S_{i}QD_{i} & \mathrm{diag}\left(Q_{N+1,1:N}\right)H^{\mathrm{BI}}\left(\sum_{j=1}^{K}W_{j}+\sum_{r=1}^{R}V_{r}\right)\\ {\left(\mathrm{diag}\left(Q_{N+1,1:N}\right)H^{\mathrm{BI}}\left(\sum_{j=1}^{K}W_{j}+\sum_{r=1}^{R}V_{r}\right)\right)}^{H} & \left(\sum_{j=1}^{K}W_{j}+\sum_{r=1}^{R}V_{r}\right) \end{array}\right]\\ & \;\;\;\;⪰0,\;\forall k. \end{array}$$
(44)$$\begin{array}{cc} {\hat{\Gamma }}_{k}= & \left[\begin{array}{cc} \sum_{i}{\bar{S}}_{i}\bar{Q}D_{i} & \mathrm{diag}\left(Q_{N+1,1:N}\right)H^{\mathrm{BI}}\left(\sum_{j\neq k}^{K}W_{j}+\sum_{r=1}^{R}V_{r}\right)\\ {\left(\mathrm{diag}\left(Q_{N+1,1:N}\right)H^{\mathrm{BI}}\left(\sum_{j\neq k}^{K}W_{j}+\sum_{r=1}^{R}V_{r}\right)\right)}^{H} & \left(\sum_{j\neq k}^{K}W_{j}+\sum_{r=1}^{R}V_{r}\right) \end{array}\right]\\ & \;\;\;\;⪰0,\;\forall k. \end{array}$$

Based on the above operation, the optimization problem (Equation 36) is transformed into the following problem:(45)$$\begin{array}{cc} \; & \max_{t,\lambda_{l},\bar{\lambda },Q}\;\;t\\ \; & s.t.\;\;{\tilde{R}}_{k}-\tau_{k,l}^{\max}\geq R_{k}^{\min},\\ \; & \;\;\;\;\;\;\left(\sum_{k=1}^{K}\mathrm{Tr}\left(\Phi_{r}W_{k}{\left(\Phi_{r}\right)}^{H}Q\right)+\sum_{m=1}^{R}\mathrm{Tr}\left(\Phi_{r}V_{m}{\left(\Phi_{r}\right)}^{H}Q\right)\right)>=t\\ \; & \;\;\;\;\;\;\hat{C3}:\;Q_{n,n}=1,Q⪰0\\ \; & \;\;\;\left[\begin{array}{cc} \lambda_{l}I_{N+M}-\hat{\Gamma } & -{\hat{\Gamma }}^{H}{\bar{g}}_{l}\\ -{\left({\hat{\Gamma }}^{H}{\bar{g}}_{l}\right)}^{H} & -\lambda_{l}{\left(\varepsilon_{l}^{E}\right)}^{2}-{\bar{g}}_{l}^{H}\hat{\Gamma }{\bar{g}}_{l}-Z_{1} \end{array}\right]⪰0\\ \; & \;\;\;\left[\begin{array}{cc} {\bar{\lambda }}_{l}I_{N+M}-{\hat{\Gamma }}_{k} & -{\hat{\Gamma }}_{k}^{H}{\bar{g}}_{l}\\ -{\left({\hat{\Gamma }}_{k}^{H}{\bar{g}}_{l}\right)}^{H} & -{\bar{\lambda }}_{l}{\left(\varepsilon_{l}^{E}\right)}^{2}-{\bar{g}}_{l}^{H}{\hat{\Gamma }}_{k}{\bar{g}}_{l}-Z_{1} \end{array}\right]⪰0\\ \; & \;\;\;\mathrm{Rank}(Q)=1 \end{array}$$

However, there is a rank-one constraint in (45) and the semidefinite relaxation method is generally used to discard the rank-one constraint; then, (45) becomes a convex optimization problem, and then uses the CVX toolbox to solve the problem. To handle the rank-one constraint, the equivalent substitution can be used to express the rank-one constraint as
(46)$$\mathrm{Rank}\left(Q\right)=1\Leftrightarrow {∥Q∥}_{☆}-{∥Q∥}_{2}=0$$

If $\forall X\in R^{M\times M}$, the inequality ${∥X∥}_{☆}=\sum_{i}\sigma_{i}\geq {∥X∥}_{2}$ holds, where $\sigma_{i}$ is the *i*-th largest singular value and the equation holds if and only if Rank(X)=1. Therefore, the penalty function method is used to put this constraint into the objective function for processing. Penalty factor μ>0 is introduced. Then, (45) can be written as
(47)$$\begin{array}{cc} \; & \max_{t,\lambda_{l},\bar{\lambda },Q}\;\;t+\frac{1}{2\mu }{(∥Q∥}_{☆}-{∥Q∥}_{2})\\ \; & s.t.\;\;{\tilde{R}}_{k}-\tau_{k,l}^{\max}\geq R_{k}^{\min},\\ \; & \;\;\;\;\;\;\left(\sum_{k=1}^{K}\mathrm{Tr}\left(\Phi_{r}W_{k}{\left(\Phi_{r}\right)}^{H}Q\right)+\sum_{m=1}^{R}\mathrm{Tr}\left(\Phi_{r}V_{m}{\left(\Phi_{r}\right)}^{H}Q\right)\right)>=t\\ \; & \;\;\;\;\;\;\hat{C3}:\;Q_{n,n}=1,Q⪰0\\ \; & \;\;\;\left[\begin{array}{cc} \lambda_{l}I_{N+M}-\hat{\Gamma } & -{\hat{\Gamma }}^{H}{\bar{g}}_{l}\\ -{\left({\hat{\Gamma }}^{H}{\bar{g}}_{l}\right)}^{H} & -\lambda_{l}{\left(\varepsilon_{l}^{E}\right)}^{2}-{\bar{g}}_{l}^{H}\hat{\Gamma }{\bar{g}}_{l}-Z_{1} \end{array}\right]⪰0\\ \; & \;\;\;\left[\begin{array}{cc} {\bar{\lambda }}_{l}I_{N+M}-{\hat{\Gamma }}_{k} & -{\hat{\Gamma }}_{k}^{H}{\bar{g}}_{l}\\ -{\left({\hat{\Gamma }}_{k}^{H}{\bar{g}}_{l}\right)}^{H} & -{\bar{\lambda }}_{l}{\left(\varepsilon_{l}^{E}\right)}^{2}-{\bar{g}}_{l}^{H}{\hat{\Gamma }}_{k}{\bar{g}}_{l}-Z_{1} \end{array}\right]⪰0 \end{array}$$
Let $Q^{☆}$ be the optimal solution of problem (45) and the penalty factor be $\mu_{☆}$. When μ→0, the limit value of $Q^{☆}$ is an optimal solution to (45). Therefore, when the penalty factor is sufficiently small, a rank-one solution can be obtained by solving (47). However, the penalty function of (47) is a non-convex DC form; then, the first-order Taylor expansion is used for processing; that is
(48)$$\frac{1}{2\mu }{∥Q∥}_{2}\geq \frac{1}{2\mu }{∥Q^{\left(i\right)}∥}_{2}+{▽}_{Q}\left(\frac{1}{2\mu }{∥Q^{\left(i\right)}∥}_{2}\right)\left(Q-Q^{\left(i\right)}\right)$$
where ${▽}_{Q}\left(\frac{1}{2\mu }{∥Q^{\left(i\right)}∥}_{2}\right)=\frac{1}{2\mu }\lambda_{\max}\left(Q\right)\lambda_{\max}^{H}\left(Q\right)$, $\lambda_{\max}\left(Q\right)$ denotes the eigenvector corresponding to the largest eigenvalue. Then, (47) can be equivalently transformed into the following problem:(49)$$\begin{array}{cc} \; & \max_{t,\lambda_{l},\bar{\lambda },Q}\;\;t+\frac{1}{2\mu }{(∥Q∥}_{☆}-{∥Q^{\left(i\right)}∥}_{2}-{▽}_{Q}\left(∥Q^{\left(i\right)}{∥}_{2}\right)\left(Q-Q^{\left(i\right)}\right))\\ \; & s.t.\;\;{\tilde{R}}_{k}-\tau_{k,l}^{\max}\geq R_{k}^{\min},\\ \; & \;\;\;\;\;\;\left(\sum_{k=1}^{K}\mathrm{Tr}\left(\Phi_{r}W_{k}{\left(\Phi_{r}\right)}^{H}Q\right)+\sum_{m=1}^{R}\mathrm{Tr}\left(\Phi_{r}V_{m}{\left(\Phi_{r}\right)}^{H}Q\right)\right)>=t\\ \; & \;\;\;\;\;\;\hat{C3}:\;Q_{n,n}=1,Q⪰0\\ \; & \;\;\;\left[\begin{array}{cc} \lambda_{l}I_{N+M}-\hat{\Gamma } & -{\hat{\Gamma }}^{H}{\bar{g}}_{l}\\ -{\left({\hat{\Gamma }}^{H}{\bar{g}}_{l}\right)}^{H} & -\lambda_{l}{\left(\varepsilon_{l}^{E}\right)}^{2}-{\bar{g}}_{l}^{H}\hat{\Gamma }{\bar{g}}_{l}-Z_{1} \end{array}\right]⪰0\\ \; & \;\;\;\left[\begin{array}{cc} {\bar{\lambda }}_{l}I_{N+M}-{\hat{\Gamma }}_{k} & -{\hat{\Gamma }}_{k}^{H}{\bar{g}}_{l}\\ -{\left({\hat{\Gamma }}_{k}^{H}{\bar{g}}_{l}\right)}^{H} & -{\bar{\lambda }}_{l}{\left(\varepsilon_{l}^{E}\right)}^{2}-{\bar{g}}_{l}^{H}{\hat{\Gamma }}_{k}{\bar{g}}_{l}-Z_{1} \end{array}\right]⪰0 \end{array}$$

Problem (49) belongs to the SDP programming, which can be solved via the CVX solver. Under the alternating optimization framework, (Equation 10) is overcome by solving problems (Equation 34) and (49) in an iterative form. The proposed robust resource allocation Algorithm 1 is summarized as follows.
**Algorithm 1** Proposed Robust Resource Allocation Algorithm1:**Input:** The channels $h_{k}$, $H^{BI}$, $h_{k}^{BU}$, $h_{l}^{IE}$, $h_{l}^{BE}$, ∀k,l;2:**Output**: $W_{k}^{☆}$, $V_{r}^{☆}$ and $q^{☆}$;3:**Initialization:**l=0, $W_{k}^{\left(n\right)}=W_{k}^{\left(0\right)}$,$V_{r}^{\left(n\right)}=V_{r}^{\left(0\right)}$, $q^{\left(0\right)}$ and a threshold ϖ;4:**Repeat**Solve (Equation 34) via CVX tool to get $W_{k}^{\left(i+1\right)}$, $V_{r}^{\left(i+1\right)}$, $t^{\left(i+1\right)}$Solve (49) via CVX tool to get $q^{\left(i+1\right)}$,i=i+1;5:**Until**$|t^{\left(i+1\right)}-t^{\left(i\right)}|\leq \varpi$.6:**end**

## 4. Simulation Results

In this section, the effectiveness of the proposed algorithm is verified by simulation. The simulation scenario is shown in Figure 1, assuming that the base station is located at (0,0), the information receivers are located in a circle with a center of $\left(x_{IU},0\right)$ with the radius equal to 1 m, the eavesdroppers are located in a circle with a center of $\left(x_{E},0\right)$ with the radius equal to 1 m, the energy receivers are located in a circle with a center of $\left(x_{R},0\right)$ with the radius equal to 1 m, and the RIS is located at $\left(x_{RIS},y_{RIS}\right)$. Define the system bandwidth and pass-loss as 10 MHz and $L\left(d\right)=T_{0}{\left(d/d_{0}\right)}^{-\alpha }$, where $T_{0}=-30$ dB denotes the pass-loss with $d_{0}=1$ m; *d* indicates the communication distance. Let the path loss factors from the base station to the RIS, the user and the eavesdropper and the energy harvesting device be $\alpha_{\mathrm{BI}}=2.2$, $\alpha_{\mathrm{BR}}=3.8$, $\alpha_{\mathrm{BU}}=3.8$, $\alpha_{\mathrm{BE}}=2.8$, respectively, while the RIS to the user, eavesdropper and energy harvesting concerning the path loss factors of the device are $\alpha_{IU}=2.4$, $\alpha_{IR}=2.2$, $\alpha_{IE}=2.4$ respectively. Define the maximum estimation error of the wiretap channel as $\varepsilon^{E}=\varepsilon_{l}^{E}$, ∀l; its normalized estimation error is expressed as $\xi =\frac{\varepsilon^{E}}{∥△g_{l}{∥}_{F}}$.

Figure 2 shows convergence performance of the max-min harvested energy among all of the energy receivers. It can be seen from the figure that the proposed algorithm in this paper reaches convergence after only a few iterations, which illustrates the proposed algorithm has good convergence. As the number of base station antennas increase, the harvested energy of the system increases. Because the number of antennas is increased, the energy beamforming gain and the signal beamforming gain are increased and the beamforming effect is improved, so that the max-min harvested energy is improved.

Figure 3 shows the relationship between the system energy efficiency and $P_{\max}$. It can be seen from Figure 3 that under the same number of RIS reflection units, the max-min harvested energy increases with the increase in the maximum transmit power $P_{\max}$. The main reason is that increasing $P_{\max}$ can increase the feasible region, thereby improving the transmission rate. We can also conclude that when the number of reflectors of the RIS increases, the harvested energy also increases. This is because adjusting the phase shift can reflect more received signals from the base station, provide more flexibility for resource allocation, improve the beam gain from the RIS to the legitimate user link and thus improve the max-min harvested energy.

Figure 4 presents the relationship between the max-min harvested energy and the number of transmit antennas. Under the same number of users, the max-min harvested energy of the system increases with the increase in the number of transmit antennas. When the number of system users increases, the max-min harvested energy among all the energy receivers is increased because the proposed algorithm maximizes the max-min harvested power of the system. As the number of users is increased, the available resource is also increased, thereby allowing harvested energy improvement.

Figure 5 presents the max-min harvested energy by the energy receiver versus receiver position. Comparison and analysis are made from the perspective of different numbers of RIS reflection elements and without RIS. It can be seen that the power collected by the energy harvesting device decreases as the distance between the receiver and the base station increases. As expected, more power can be collected with RIS than without RIS, especially when the number of reflectors is large. Because the use of the RIS adds an additional reflection link, the reception gain of the energy receiver is enhanced.

To further verify the performance of the proposed algorithm, the max-min harvested energy robust algorithm based on imperfect CSI is denoted as the proposed algorithm; the max-min harvested energy algorithm based on perfect CSI is defined as “proposed algorithm (perfect CSI)”; the max-min harvested energy robust algorithm based on imperfect CSI with random phase shift is denoted as “Random Phase Shift”; the max-min harvested energy algorithm without considering IRS is defined as “Without RIS”.

Figure 6 shows the relationship between the max-min harvested energy and the user secure rate threshold. With the increase in $R_{k}^{\min}$, the max-min harvested energy of different algorithms gradually decreases. As $R_{k}^{\min}$ increases, the system must allocate additional power to satisfy the secrecy rate constraint, resulting in a decrease in fairness harvested energy. The fairness-aware harvested energy of the proposed robust algorithm with imperfect CSI is significantly higher than that of the conventional algorithm without RIS, which indicates that the proposed algorithm can utilize the spatial degrees of freedom to provide security more effectively than the algorithm without RIS, even in the presence of uncertain channel parameters. However, the max-min harvested energy of the proposed algorithm is slightly lower than that of the perfect CSI algorithm because the base station and RIS can not allocate beams accurately due to the existence of channel uncertainty parameters. As can be observed, for the proposed scheme, the average max-min harvested energy decreases as the quality of the CSI degrades. In particular, the worse the quality of the estimated CSI, the more difficult it is for the base station to perform accurate beamforming and efficient AN jamming, which results in a lower achievable fairness harvested energy.

## 5. Conclusions

In order to improve the ability of the RIS-assisted MISO system to overcome channel uncertainty and user information leakage, this paper proposes a resource allocation algorithm based on the bounded channel uncertainty model to maximize the minimum harvested energy of the system. Considering the necessary physical constraints and user transmission quality constraints, a multi-variable coupled robust harvested energy optimization problem is constructed. The original problem is transformed into a deterministic problem by using S-procedure methods; at the same time, the problem is transformed into a convex optimization problem by using the alternating optimization method. The simulation results verify the superiority of the algorithm in this paper.

## Figures and Tables

**Figure 1 sensors-22-08274-f001:**
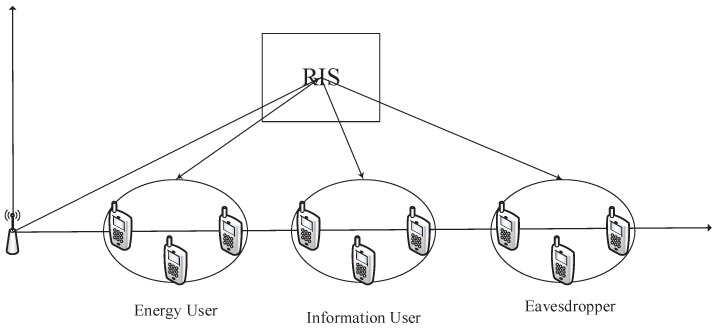
Simulation scenario.

**Figure 2 sensors-22-08274-f002:**
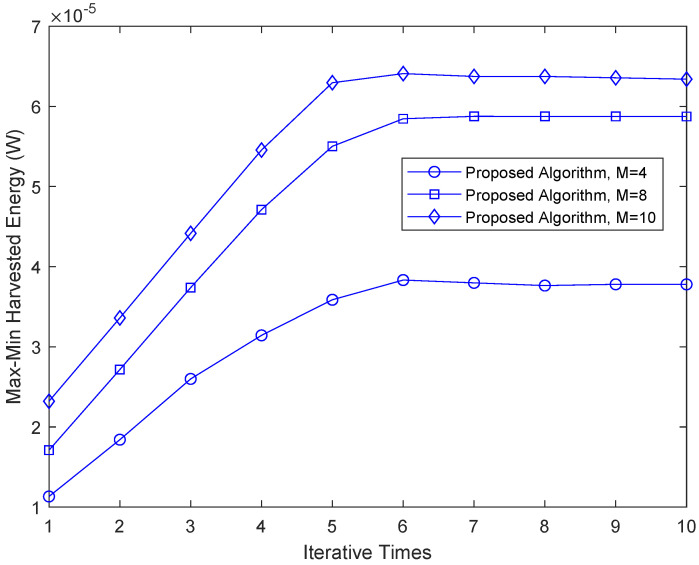
Convergence performance of the proposed algorithm under different numbers of reflecting elements.

**Figure 3 sensors-22-08274-f003:**
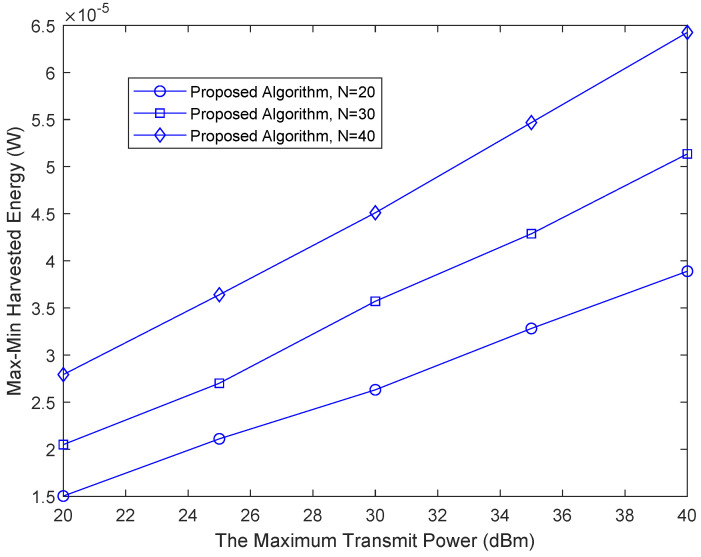
Relationship between max-min harvested energy and the maximum transmit power of base station.

**Figure 4 sensors-22-08274-f004:**
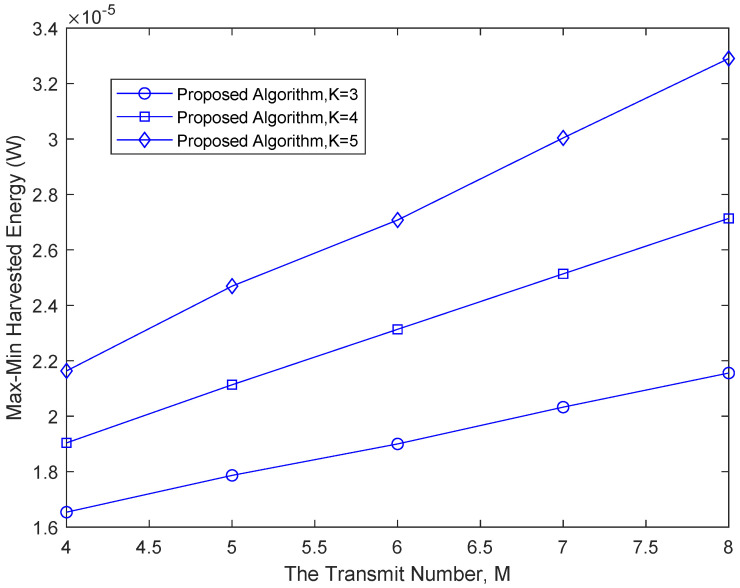
The relationship between the max-min harvested energy and the number of transmit antennas.

**Figure 5 sensors-22-08274-f005:**
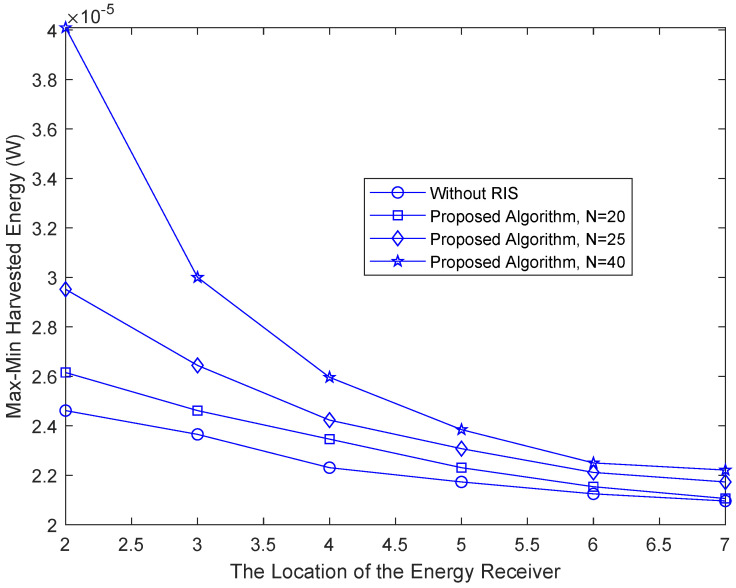
Max-min harvested energy by an energy receiver versus receiver location.

**Figure 6 sensors-22-08274-f006:**
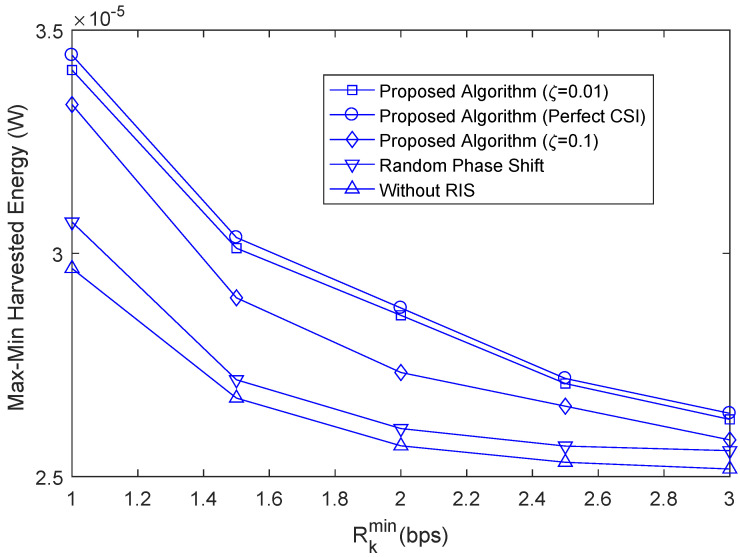
The relationship between the max-min harvested energy and the user secure rate threshold.

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
