# Peer review of "Robust Secure Resource Allocation for RIS-Aided SWIPT Communication Systems"

_sensors, 2022, doi:10.3390/s22218274_

Round 1
Reviewer 1 Report
1. How is resource allocation related with security in RIS? The paper did not clarify that.
2. The swipt is not clearly described in this paper although it is mentioned in the title and references.
3. Please explain explicitly the formulated problem in (11).
4. In the simulation results, only the harvested energy is validated. How about the robustness and security?
5. More new literatures on resource management can be included:
(1) X. Xu, S. Zhang, F. Gao and J. Wang, "Sparse Bayesian Learning Based Channel Extrapolation for RIS Assisted MIMO-OFDM," in IEEE Transactions on Communications, vol. 70, no. 8, pp. 5498-5513, Aug. 2022
(2) C. Wang, D. Deng, L. Xu and W. Wang, "Resource Scheduling Based on Deep Reinforcement Learning in UAV Assisted Emergency Communication Networks," in IEEE Transactions on Communications, vol. 70, no. 6, pp. 3834-3848, June 2022
Author Response
Response to the Editor and Reviewers
Dear Editor and Reviewers,
We appreciate the opportunity of presenting our work for your consideration. We thank the valuable time and effort of the Editor and anonymous Reviewers towards improving and clarifying the presentation of our ideas. After being carefully and thoroughly revised, the manuscript has undergone several modifications.
In the revised manuscript, we have highlighted the main modifications in red for your convenience. Next, we present detailed answers for each of the Reviewers’ comments.
Comment 1: How is resource allocation related with security in RIS? The paper did not clarify that.
Response: Thank you very much for your constructive comments and suggestions. Motivated by the aforementioned observations, we consider a robust and secure IRS-assisted multiuser MISO-SWIPT downlink wireless system in the presence of multiple single-antenna potential eavesdroppers. . For security provisioning in our considered systems, we adopt the worst-case assumption on the eavesdropping capabilities of potential eavesdroppers. Therefore, the channel capacity between the AP and the l-th potential eavesdropper for decoding the signal of the k-th legitimate user is given by (7) in our revised manuscript.
Comment 2: The swipt is not clearly described in this paper although it is mentioned in the title and references.
Response: Thank you very much for your constructive comments and suggestions. In our revised manuscript, we have provided more description about SWIPT.
Comment 3: Please explain explicitly the formulated problem in (11).
Response: Many thanks for this comment. The objective function in (11) maximizes the minimum value of all the possible sum-rate due to CSI errors. Constraint C1 represents the minimum security rate constraint of the legal user k. Constraint C2 specifies the maximum transmit power constraint of the base station. Constraint C3 represents the phase shift constraint of RIS.
Comment 4: In the simulation results, only the harvested energy is validated. How about the robustness and security?
Response: Thank you very much for this useful comment. In our updated manuscript, we have added some new simulation results to show the robustness and security, for example:
“Figure \ref{FIG116} shows the relationship between the Max-min harvested energy and the user secure rate threshold. With the increase of $R_{k}^{\min}$, the max-min harvested energy of different algorithms gradually decreases. As $R_{k}^{\min}$ increases, the system must allocation additional power to satisfy the secrecy rate constraint, resulting in a decrease in fairness harvested energy. The fairness-aware-harvested-energy of the proposed robust algorithm with imperfect CSI is significantly higher than that of the conventional algorithm without RIS, which indicates that the proposed algorithm can utilize the spatial degrees of freedom to provide security more effectively than the algorithm without RIS, even in the presence of uncertain channel parameters. However, the max-min harvested energy of the proposed algorithm is slightly lower than that of the perfect CSI algorithm because the base station and RIS can not allocate beams accurately due to the existence of channel uncertainty parameters. As can be observed, for the proposed scheme, the average max-min harvested energy decrease as the quality of the CSI degrades. In particular, the worse the quality of the estimated CSI is, the more difficult it is for the base station to perform accurate beamforming and efficient AN jamming, which results in a lower achievable fairness harvested energy.”
Figure 6
Comment 5: More new literatures on resource management can be included:
(1) X. Xu, S. Zhang, F. Gao and J. Wang, "Sparse Bayesian Learning Based Channel Extrapolation for RIS Assisted MIMO-OFDM," in IEEE Transactions on Communications, vol. 70, no. 8, pp. 5498-5513, Aug. 2022
(2) C. Wang, D. Deng, L. Xu and W. Wang, "Resource Scheduling Based on Deep Reinforcement Learning in UAV Assisted Emergency Communication Networks," in IEEE Transactions on Communications, vol. 70, no. 6, pp. 3834-3848, June 2022.
Response: Thanks for the comment. In our revised manuscript, we have added the two literatures as the reference.

Reviewer 2 Report
The paper tackles an area which is recent and provides an adequate addition. However, the paper needs to improve both technically and from the organization and writing aspects. Below some of the points that can help improve the paper are mentioned.
1. In eq(1), dimensions of s_k and s_r are not defined.
2. Line 92>> denominator; please see if the double bar, ie., II shall be removed with |.
3. Equation (10) >> C2; there is not bound or equality included so it shall be corrected. Also, 106-101 shall be made as per the given equation. Presently, there are several mistakes.
4. Line 121 includes several typographical errors.
5. Since W_k=w_k w_k^H, hence the rank will always be 1. So I don’t know why have you included this as a constraint in (11).
6. Inconsistency in writing inequalities in (12) and (14).
7. Please find a better way to express (21)-(24), presently it is not readable.
8. Equation (35) needs corrections.
9. Algorithm 1 shall start by mentioning first what are inputs and outputs. Output is mentioned in line 4, so move it to line 2 after defining the input to the algorithm.
10. Figure 2, the scale of harvested energy is 10^-5 Watts. Is it any good practically. What are the benchmarks used in literature for such.
Author Response
Dear Editor and Reviewers:
We would like to thank you for handling the review process of our paper. We are also indebted to you and the reviewers for many helpful comments. Following your suggestions, we have updated the original manuscript and resubmitted a revised version. To enhance the legibility of this response letter, all the reviewers’ comments are typeset in italic font and our responses are written in plain font. Rephrased sentences are typeset in yellow.
Comment 1: In eq(1), dimensions of s_k and s_r are not defined.
Response: Thank you very much for this comment. In our revised manuscript, we have defined s_{k} and s_{r} in eq(1).
Comment 2: Line 92>> denominator; please see if the double bar, ie., II shall be removed with |.
Response: Sorry for this mistake. We have removed one |, the detail can be found in our revised manuscript.
Comment 3: Equation (10) >> C2; there is not bound or equality included so it shall be corrected. Also, 106-101 shall be made as per the given equation. Presently, there are several mistakes.
Response: Many thanks for the useful comment. In our revised manuscript, we have carefully re-checked all the expressions.
Comment 4: Line 121 includes several typographical errors.
Response: Thank you very much for pointing these errors. These errors is corrected in our updated manuscript.
Comment 5: Since W_k=w_k w_k^H, hence the rank will always be 1. So I don’t know why have you included this as a constraint in (11).
Response: Many thanks for this valuable comment. Since W_k=w_k w_k^H, hence the rank will always be 1, which leads the original optimization problem is equivalent to the optimization problem (11). The rank-1 constraint makes the problem (11) difficult to solve. Thus, we apply SDR method to relax the constraints. After the W_k is obtained from the relaxed optimization problem (34), we can see that W_{k} is not always satisfied rank-1 constraint. If the rank of W_{k} equals to 1, we can get w_{k} by performing eigenvalue decomposition. Otherwise, the Gaussian randomization can be used for recovering the approximate w_{k}.
Comment 6: Inconsistency in writing inequalities in (12) and (14).
Response: Thank you very much for this comment. We have revised the inequalities, the details can be found in our revised manuscript.
Comment 7: Please find a better way to express (21)-(24), presently it is not readable.
Response: Sorry for your poor reading experience. In our updated manuscript, we have adjusted the layout of the formulations.
Comment 8: Equation (35) needs corrections.
Response: Thank you very much for this comment. We have corrected this error in our updated manuscript.
Comment 9: Algorithm 1 shall start by mentioning first what are inputs and outputs. Output is mentioned in line 4, so move it to line 2 after defining the input to the algorithm.
Response: Many thanks for your valuable suggestions. Following your suggestion, we have added the input parameters as well as Output is moved to line 2 defining the input to the algorithm.
Comment 10: Figure 2, the scale of harvested energy is 10^-5 Watts. Is it any good practically. What are the benchmarks used in literature for such.
Response : Thank you very much for this comment. Due to the large-scale path-loss and the poor energy harvesting efficiency, the harvested energy is 10^(-5)Watts, the related references can be found in [15] and [19].

Round 2
Reviewer 1 Report
The author have well addressed my concerns. I don't have further comments.
Author Response
Comment 1: The author has well addressed my concerns. I don't have further comments.
Response: Thank you very much for your constructive comments and suggestions.

Reviewer 2 Report
Authors' have addressed my comments well. Paper can be accepted for publication, just fix issues in line 146.
Author Response
Comment 1: Authors' have addressed my comments well. Paper can be accepted for publication, just fix issues in line 146.
Response: Thank you very much for this comment. In our revised manuscript, we have revised the number of the expression in line 146 .
